# Tempo: Accelerating Transformer-Based Model Training through Memory Footprint Reduction

**Muralidhar Andoorveedu[1], Zhanda Zhu[2,3], Bojian Zheng[1,3], Gennady Pekhimenko[1,3]**

[1]University of Toronto, Toronto, Canada
[2]Shanghai Jiao Tong University, Shanghai, China
[3]Vector Institute, Toronto, Canada
{andoorve, zhanda, bojian, pekhimenko}@cs.toronto.edu

## Abstract

Training deep learning models can be computationally expensive. Prior works have shown that increasing the batch size can potentially lead to better overall throughput. However, the batch size is frequently limited by the accelerator memory capacity due to the activations/feature maps stored for the training backward pass, as larger batch sizes require larger feature maps to be stored. Transformer-based models, which have recently seen a surge in popularity due to their good performance and applicability to a variety of tasks, have a similar problem. To remedy this issue, we propose *Tempo*, a new approach to efficiently use accelerator (e.g., GPU) memory resources for training Transformer-based models. Our approach provides drop-in replacements for the GELU, LayerNorm, and Attention layers, reducing the memory usage and ultimately leading to more efficient training. We implement *Tempo* and evaluate the throughput, memory usage, and accuracy/loss on the $\text{BERT}_{LARGE}$ pre-training task. We demonstrate that *Tempo* enables up to $2\times$ higher batch sizes and 16% higher training throughput over the state-of-the-art baseline. We also evaluate *Tempo* on GPT2 and RoBERTa models, showing 19% and 26% speedup over the baseline.

## 1 Introduction

Transformer-based models such as BERT [12] and GPT-2 [49] have found success in numerous general natural language processing tasks including question answering [51], paraphrasing [13], natural language inference [68], and even areas outside language tasks such as image recognition [14]. However, training such models can be highly expensive in terms of time, monetary resources and carbon footprint [24, 60]. For instance, the pre-training of $\text{BERT}_{LARGE}$ takes 4 days to complete on 16 Cloud TPUs (64 TPU chips total) [12], which costs about $10,000 [56]. Training a more recent Transformer-based model, GPT-3, has an even more astonishing price tag - $12 million[66]. Hence, even a small decrease in the end-to-end training time of Transformer-based models matters.

Although there has been significant progress made in accelerating Transformers using specialized hardware (e.g., Google TPUs [30], NVIDIA Tensor Cores [39]) in the past few years, a fundamental issue with Transformer-based models is that they are limited by the memory capacity of hardware accelerators. For example, even a batch size of 1 does not fit into a modern GPU with 12GB of memory when training BERT with sequence length 512 [15]. Reducing memory footprint [48, 8, 52] is a viable option to allow larger batch training, leading to better hardware utilization and ultimately improved training throughput [73].

Many existing approaches to memory footprint reduction (e.g., offloading [52, 65, 48], checkpointing [8, 73, 33, 28], and data compression/encoding [26, 6]) either have high computational overhead or do not apply to Transformer-based models directly. Prior approaches fall into two main categories,

neither of which are satisfactory for the Transformer-based model case. First, these techniques may be too general [48, 6, 50, 33, 28] to utilize the specifics of Transformer-based models well, such as the multi-headed attention mechanism used in Transformers [63], or optimization opportunities available in specific layers such as the LayerNorm [4] layer. For example, although checkpointing [8, 28] can significantly enlarge batch size, it also brings high overhead (e.g., 30% performance degradation observed in some prior works [8]). Second, if prior works are specific, they focus on other types of models/layers with ideas not being applicable to Transformers. For example, Gist and In-Place ABN deal with CNNs [26, 53].

In our work, we demonstrate that low overhead memory footprint reduction can lead to a positive improvement in throughput. In addition, unlike prior works which do not leverage the specifics of Transformer-based models, we propose a new approach specifically tailored for Transformer-based models, called *Tempo*. This approach includes three new techniques: (i) In-place GELU, (ii) In-place LayerNorm, and (iii) Sub-Layer Dropout Recomputation. In-place GELU and In-place LayerNorm both use alternative derivations for the computation of the backward passes of these layers. These derivations allow some activations that are normally retained during the forward pass (to be later used in the backward pass) to be discarded, leading to a more memory-efficient implementation. Sub-Layer Dropout Recomputation discards activations within the high memory footprint attention mechanism during the forward pass, then recomputes these during the backward pass without recomputing extra unnecessary tensors. *Tempo* is able to increase the training throughput with larger batch sizes by reducing the total memory footprint of the models during training. To our best knowledge, this is the first work to explore memory footprint optimizations specifically for Transformer-based layers that show not just footprint reduction, but the actual increase in throughput using the extra memory savings. Tempo reduces the memory footprint of training Transformer-based models by targeting a major part of the total footprint – the *activation memory* [74] (the saved feature maps during the forward pass of the model that are required for backpropagation [54]). All the proposed techniques provide a large memory footprint reduction with very low throughput degradation (as low as 1%). Our results show up to $2\times$ improvement in batch size for $\text{BERT}_{LARGE}$ pre-training at a sequence length of 512 on modern GPUs while increasing the training throughput by up to 16% .

## 2 Background and Motivation

### 2.1 Memory Footprint of BERT

BERT [12] is a popular natural language processing model that is based on the Transformer architecture [63]. The model has been successfully applied to a variety of tasks such as question answering (SQuAD [51]), paraphrasing (MRPC [13]), natural language inference (MNLI [68]), and others [57, 72] through a two step training process. The training process entails first training on a general unlabelled data set (*pre-training*) [12]. The faster second part of the training process (*fine-tuning*) takes the parameter weights produced by the pre-training section and further trains on a downstream task such as question answering [51] or sentiment analysis [57] which it accomplishes through the addition of a specialized output layer [12].

The BERT architecture allows for multiple different configurations depending on model hyperparameters selected, some being derived from the original Transformers paper; these include the hidden layer size ($H$), sequence length ($S$), number of attention heads ($A$) and number of layers ($L$).

In the context of this work, we point out some of the relevant parts of the model and their activation memory footprint with respect to these hyperparameters referring to Figure 1.

① At this point, where attention [63] is calculated we observe that the size of each of the feature maps goes as $\mathcal{O}(S^2)$– there are a variety of previous techniques and models that have been explored in the literature to deal with this problem [61]. Additionally, at this point note that we store three feature maps of size $[B \times A \times S^2]$. Calculations based on Figure 1 at the $\text{BERT}_{BASE}$ parameters show that at a sequence length of 512 these three feature maps **account for** $56\%$ **of the encoder layer activation memory.**

② At these two points, we store the input to the two LayerNorm layers of size $[B \times S \times H]$

③ Here a GELU [21] layer is used as the activation function for the preceding fully-connected layer of size $[B \times S \times 4H]$. The activation memory for this function stores **almost** $17\%$ **of the total layer activation memory of** $\text{BERT}_{BASE}$ at a sequence length of 128.

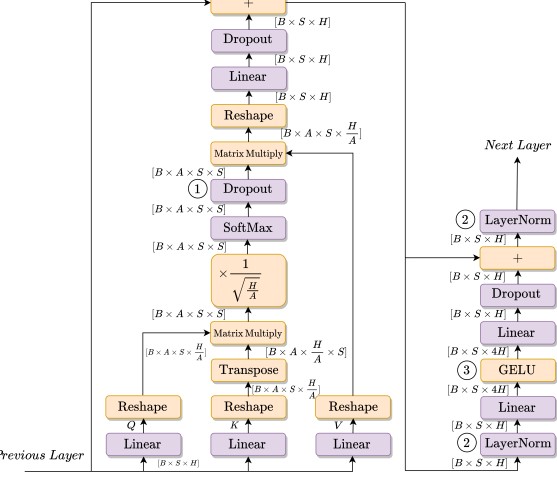

Figure 1: A diagram of a single Transformer encoder [63] layer used in BERT [12]. This is based on the Huggingface implementation of BERT [69]. As in the BERT paper, $A$ represents the number of attention heads, and $H$ represents the hidden size. We represent the batch size by $B$ and the sequence length by $S$. Sizes of intermediate tensors (both retained activations and unretained intermediates) are annotated.

## 2.2 Why Activation Memory Matters

As iterated in previous works [52, 26, 48, 73, 6] there are multiple benefits to reducing the memory footprint of models. First, it allows for larger models which can positively affect the model's performance on downstream tasks [12]. Second, memory footprint reduction can allow for a larger batch size. This, in turn, could lead to better utilization of the GPU hardware [17], increasing the overall throughput [73]. In order to verify this possibility for Transformer-based models, we conduct our own experiments using Huggingface's BERT implementation [69] to train $\text{BERT}_{LARGE}$ on the MRPC [13] fine-tuning task. Figure 2 shows the throughput on this task for sequence lengths of 128 and 512. From the figure, we conclude that there is a steady improvement in batch size when the sequence length is 128. This is also the case when the sequence length is 512, however, in this situation the trend ends more abruptly as the memory consumption of the model exceeds the GPU memory capacity, showing a clear opportunity to take advantage of memory footprint reduction.

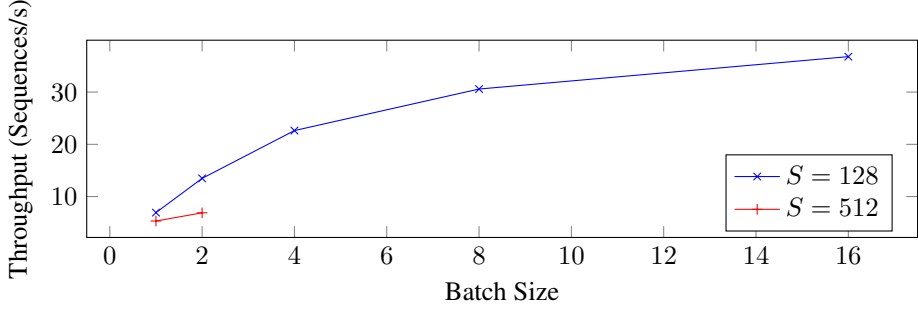

Figure 2: Plots of throughput (sequences/s) vs batch size for $\text{BERT}_{LARGE}$ [12] fine-tuning on the MRPC [13] task at sequence lengths 128 and 512 on four 2080Ti [40] GPUs. The maximum batch sizes are respectively 16 and 2.

We note that previous works on Transformer-based models show that although the model parameters contribute to the memory footprint, the main memory capacity consumer during training is actually the activation feature maps [74, 28, 8, 48, 26, 33, 6]. In addition, the majority of this activation memory will be used in each of the BERT Transformer encoder layers. Profiling the Huggingface $\text{BERT}_{BASE}$ implementation [69] on the MRPC [13] fine-tuning task at a batch size of 32 and

sequence length of 128 shows that 66% of the total memory is taken up by these encoder activations. More details on this are shown in Appendix A.

## 2.3  Key Prior Works

There are three major prior techniques used in training memory footprint reduction of deep learning models. The first of these is *Checkpointing* [8, 33, 28, 73]. This technique involves discarding certain feature maps in the forward pass while retaining others. Later, in the backward pass, these discarded feature maps may be recomputed from the retained feature maps, and thus used in the computation of the gradients. The second technique is *Offloading* [48, 52, 65]. In this case, the main idea involves taking feature maps that would be stored in the GPU memory, and instead offloading them to the CPU memory. These techniques can also involve *pre-fetching* tensors from the CPU memory in anticipation of their use. Offloading suffers from a dependence on system variables such as the communication channel bandwidth [52, 48]. It also requires extensive engineering effort to avoid high overhead [6]. Finally, *Compression/encoding*; this can be divided into two different categories, lossless and lossy [26, 6]. However, the fundamental idea is to compress, or reduce the space taken up by feature maps in the forward pass, then decompress it for use in the backward pass.

These techniques are usually largely orthogonal to one another as was shown in prior works where both offloading and checkpointing are used simultaneously [48, 65]. We expand on these techniques in Appendix C.

## 2.4  Why Tempo?

Although the techniques in the previous section show good performance on a variety of models, they suffer from a variety of issues. Checkpointing's scope is often too broad to consider certain layer-specific optimizations and alternative derivations that can provide lower overhead [53]. Furthermore, overhead can be high (as much as 30%) [8]. Offloading can be system- dependent and requires significant engineering effort, while compression can be lossy or not applicable to the Transformer case. Hence, there is a clear need for a deeper look at activation memory optimizations for Transformer-based neural networks in particular. To our best knowledge, our work is the first to explore such optimizations tuned to improving the throughput of Transformer-based models. Table 1 shows a summary comparison of Tempo and various other techniques, with the major points that differentiates our technique from prior work.

| Feature | Capuchin | Checkmate | ActNN | Gist | Tempo |
|---|---|---|---|---|---|
| Layer-Specific | ✗ | ✗ | ✗ | ✓ | ✓ |
| Transformer-Specific | ✗ | ✗ | ✗ | ✗ | ✓ |
| Lossless | ✓ | ✓ | ✗ | $\sim^1$ | $\sim^2$ |
| Drop-In Layer Replacement | ✗ | ✗ | ✓ | ✓ | ✓ |
| Online | ✓ | ✗ | ✓ | ✓ | ✓ |

Table 1: Comparison between Tempo and Capuchin [48], Checkmate [28], ActNN [6], and Gist [26].

## 3  Tempo: Key Ideas

We now present the major ideas that lays behind the design of Tempo: (1) **In-place GELU**, (2) **In-place LayerNorm**, and (3) **Sub-Layer Dropout Recomputation**. The major theme behind all of these ideas is to compute the backward pass as normal, while *using less storage* to do so. To this end, In-place GELU and In-place LayerNorm compute the output of each layer *in-place*; instead using the output activation to compute the gradient. Sub-Layer Dropout Recomputation also discards the output, and through a closer look at the structure of the Dropout layer is able to recompute the output without excessive recomputation. We strongly suggest reading Appendix E for the implementation details. We also add in this appendix a new optimization of softmax that we use that further reduces memory [18].

---

[1]Some of the Gist [26] optimizations are lossy.

[2]Accuracy of our lossy optimization is tunable, offering a flexible tradeoff between the accuracy and the hardware cost.

## 3.1 In-place GELU

The GELU layer is used as an activation function for the feed-forward section of the BERT layer (③ in Figure 1) [12]. A plot of this function is shown in Figure 3a. Referring to the baseline in Figure 3b, note that both $X$ and $Y$ are stored for the backward pass. $Y$ is needed for the downstream fully connected layer, while $X$ is stored for the GELU layer itself [46]. Prior work has demonstrated that certain activation functions such as ReLU may be computed in-place [26]. This can be done without affecting the calculation of the backward pass. If we were able to compute the GELU function in-place, potentially by recovering the input from the output on the backward pass, we could save the storage required for $X$. However, this is impossible to do directly. A key observation to make with respect to the GELU function is that it is not bijective – hence there is no function that will be able to compute the input from the output without additional information.

However, we observe that the GELU function is both continuous and has only one extremum, a minimum value at $x \approx -0.75179$ as can be seen in Figure 3a. Notably, this implies that just one additional piece of information: which **side of the minimum** the input originates from, allows us to compute the inverse of the GELU. This is because on each side of the minimum the function is one-to-one, and hence the input is recoverable from the output in *each section*. Based on this key observation, we can discard the input, and simply retain the output of the GELU, as well as the additional information on whether the input is greater than or equal to the value at which the minimum occurs. Figure 3b illustrates the difference between our method and the baseline.

In order to execute this efficiently on a real system, we note that the original derivative in terms of the input can be composed with the function inverse in order to create a composite kernel. This kernel consists of a polynomial approximation of this composite function, the approximation being necessary since GELU is transcendental, and therefore the inverse cannot be solved in terms of elementary functions [58]. Further details are discussed in Appendix E.

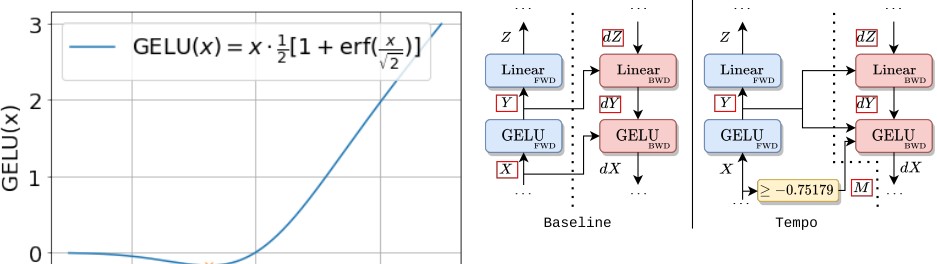

Figure 3(b): Saved feature maps between the baseline and Tempo. Note that our method only saves a 8-bit mask[3] that denotes whether the input is greater or less than the minimum value, instead of the full 32-bit input feature map.

Figure 3(a): A plot of the GELU[21] function near the origin, along with the marked minimum point.

## 3.2 In-place LayerNorm

The LayerNorm layer is used at multiple points in the Transformer encoder layer [63], which we denote by ② in Figure 1. Usually, the gradient computation of LayerNorm relies on the gradient input from the next layer, as well as the input feature map which is stashed for this computation [46].

Similar to GELU, we are able to derive an expression for the gradient of the LayerNorm layer *as a function of its output*. In this context, the output of LayerNorm must be stashed to compute the gradient of the successive fully connected layer anyways. Using this approach, the memory footprint overhead of LayerNorm is just the intermediate mean and variance computed in the forward pass. The full derivation is presented in Appendix E which is extended from the treatment of BatchNorm in [53].

**Comparison with Checkpointing**: Note that although In-place GELU requires more memory compared to recomputing $Y$ from $X$, it will have increased overhead due to the recomputation.

---

[3]Pytorch boolean masks use 8-bits per value [46]. Masks can also be implemented as 1-bit manually but this brings extra overhead due to unpacking and packing bit tensors.

Additionally, our technique is orthogonal to conventional Checkpointing, as it could take advantage of the fact that no recomputation is required for the input $X$ for both In-place GELU and In-place LayerNorm.

## 3.3 Sub-Layer Dropout Recomputation

In this section we explore the idea of sub-layer granularity checkpointing, or partial recomputation applied to the Dropout layer [59] found in ① in Figure 1. The function of a dropout layer is to set the output of $p\%$ of entries in the incoming feature map to zero ("drop" the outputs) and then scale the remaining outputs by the factor $\frac{1}{1-p}$, which makes the network less sensitive to any output of the preceding feature map, thereby making it more robust [59].

We define sub-layer recomputation as a technique where recomputation of only *some* of the feature maps is necessary for the backward pass that may be produced by a given layer's output. We observe that better recomputation strategies are possible if we carefully deconstruct layers in the case where they store multiple outputs. This observation can be directly applied to the Dropout layer. In the computation of Dropout, both a mask (which records the entries which are set to zero in implementations of Dropout [46, 7]) and output are produced. If a layer-based checkpointing implementation [16] was used, it would cause both the mask and the output to be recomputed in the backward pass if the layer is checkpointed, thus requiring higher overhead. However, we notice that nothing precludes us from simply doing only one of these recomputations. Storing the mask would only reduce the recomputation (including memory transfer) time, while the fact that the mask itself only has Boolean values allows us to keep most of the memory benefit of recomputation. In this way, we can save the storage required for the output at the critical $\mathcal{O}(S^2)$ Attention section (③ in Figure 1) for the cost of a simple mask multiply. This technique is illustrated in Figure 4.

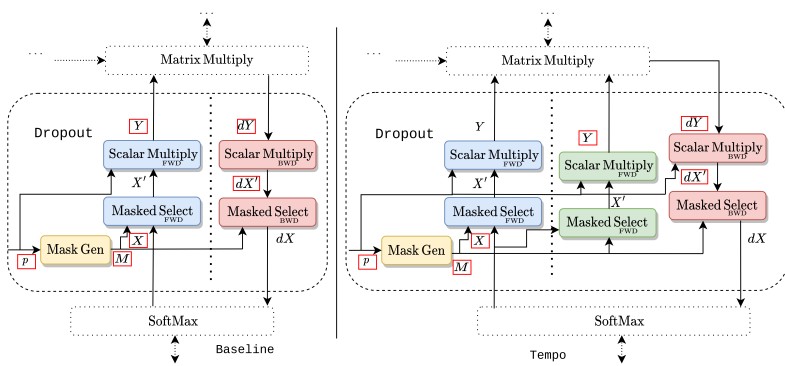

Figure 4: Comparison of dropout implementation between the baseline, and our method. Note that we only save the mask, and recompute the other output. The representation on the left is not an exact copy of the PyTorch implementation, rather it is an illustrative representation.

## 3.4 Other Engineering Optimizations

We note that PyTorch uses a memory-inefficient implementation of the softmax function which retains both the input and output of the function for the backward pass [46]. Instead, only the output is necessary. This optimization has also previously been implemented as part of some models in the Huggingface library [18]. We use this optimization as well in our implementation of the attention mechanism to further reduce the activation memory pressure.

# 4 Evaluation

## 4.1 Methodology

**Infrastructure** Our main test setup consists of 4 NVIDIA RTX 2080 Ti GPUs [40], each with 11 GB of memory connected over PCIe v3 [47]. We also use an Amazon Web Services p3.8xlarge [3] instance consisting of 4 NVIDIA Tesla V100 GPUs [39] each with 16 GB of memory connected

using NVLink [42]. For our ablation studies, we employ a system with an NVIDIA A100 GPU [43] with 40 GB of memory. We summarize the detailed setup in Appendix G.

**Applications**  We evaluate our work using both the BERT pre-training and fine-tuning tasks [12]. For pre-training, we employ the NVIDIA DeepLearningExamples library [41] with the English Wikipedia dataset [67]. We perform the training in two phases, the first (i.e., longer) phase at a sequence length of 128, and the second (i.e., shorter) phase at a sequence length of 512 [12, 41]. For throughput and memory experiments, we use the $BERT_{LARGE}$ configuration. For our fine-tuning task, we use the MRPC [13] paraphrasing task on $BERT_{BASE}$ using the Huggingface library [69].

For our ablation studies, we also train both RoBERTa [34] and GPT2[49]. For the evaluation of RoBERTa, we use the Fairseq library [44], while GPT2 uses the Huggingface GPT2 model [69]. Both of these models use the WikiText Dataset for evaluation [35].

**Metrics**  The first metric we focus on is the total *memory footprint* of our method compared to the baselines. There are two ways to look at this metric. First, we compare the maximum batch size possible for each method. We compare this across sequence lengths of 128 and 512 [5] on $BERT_{LARGE}$ for both 2080Ti and V100 GPUs. Second, we compare the total memory used by PyTorch at a given commonly used batch size for the same parameters. The second metric we use is the *throughput* for which we count the total number of sequences per second processed. Finally, we provide a comparison between our method and the baseline method on $BERT_{BASE}$ pre-training in order to compare the loss curves and show the change due to our lossy optimizations. We also provide fine-tuning curves on the MRPC [13] task, training for 10 epochs to ensure no significant accuracy deviations.

Our ablation studies only use the throughput metric.

## 4.2 Results

We use two major baselines. The first baseline is the NVIDIA $BERT_{LARGE}$ model [41], with no memory footprint techniques applied which we refer to as the *Baseline*. The second one is the same model, with the default checkpointing applied, based on the PyTorch implementation, applied at the input of each Transformer encoder layer [46, 41] and is similar to the Huggingface implementation [69]. We refer to this baseline as *Checkpoint*. We refer to our method that uses In-Place GELU, In-Place LayerNorm, Sub-Layer Dropout Recomputation, and the softmax engineering optimization as *Tempo*.

**Impact on Memory Footprint**  Table 2 shows the maximum batch size and memory consumed at a fixed batch size for all three methods. Additionally, the total memory used at a batch size of 15 at a sequence length of 128 is 11.3 GB, 8.3 GB and 9.2 GB respectively for *Baseline*, *Checkpoint*, and *Tempo*. From this, we conclude that *Checkpoint* reduces the memory footprint to a much higher degree than both *Baseline* and *Tempo*. This is expected, as *Checkpoint* discards most of the feature maps to be recomputed [69, 41] no matter the performance cost. *Tempo* still provides a significant increase in batch size over *Baseline* at the sequence length of 512 – **we see 2× and 1.5× larger batches over *Baseline*** for the 2080 Ti and V100 respectively but, as the next section shows, with much better throughput.

| Technique | Sequence Length | Batch Size | Technique | Sequence Length | Batch Size |
|---|---|---|---|---|---|
| Baseline | 128 | 15 | Baseline | 128 | 28 |
| Baseline | 512 | 1 | Baseline | 512 | 4 |
| Checkpoint | 128 | 50 | Checkpoint | 128 | 96 |
| Checkpoint | 512 | 4 | Checkpoint | 512 | 18 |
| Tempo | 128 | 24 | Tempo | 128 | 41 |
| Tempo | 512 | 2 | Tempo | 512 | 7 |

Table 2: The maximum batch size on both 2080 Ti (left) and V100 (right) for $BERT_{LARGE}$.

**Impact on Throughput**  Figure 5 illustrates our main results with respect to throughput. From the figure, we can see that *Tempo* outperforms both *Checkpoint* and *Baseline* across **both sequence**

---

[5]These are the sequence lengths of Phase 1 and Phase 2 of pre-training [12, 41].

**lengths** and **across different hardware setups**. We observe an improvement of **16%** over *Baseline* on the 2080 Ti at a sequence length of 512. At these settings, we also have an improvement of 8% over *Checkpoint*. We also observe up to **27%** over *Checkpoint* on the V100 at a sequence length of 512, which also corresponds to a 5% improvement over *Baseline*. This is despite the fact that *Checkpoint* uses the largest batch size as per Table 2. This is because *Checkpoint* stores feature maps at the beginning of each Transformer encoder layer, and recomputes these layers [69, 41]. Hence, an increased batch size also means more recomputation. In contrast, *Tempo* is able to decrease the total memory footprint, and then convert this decrease into a substantial performance improvement over the *Baseline* due to the use of only low overhead mechanisms.

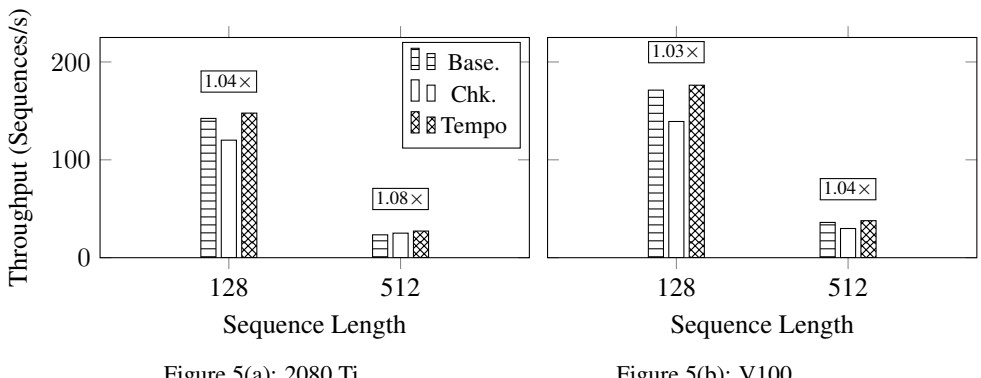

Figure 5(a): 2080 Ti

Figure 5(b): V100

Figure 5: Throughput experiments at the maximum batch size annotated with the speedup *over the best baseline*.

**Impact on Loss and Accuracy**    We pre-train $\text{BERT}_{BASE}$ to ensure that our model's loss curve is not affected by approximate optimizations (e.g., In-Place GELU). Figure 6a shows the loss curve of phase 1 of $\text{BERT}_{BASE}$ pre-training [12]. We observe almost complete overlap in the loss curves with no more than a *0.5%* difference between *Tempo* and the baseline at the endpoint. We conclude that within that margin of error our method is satisfactory.

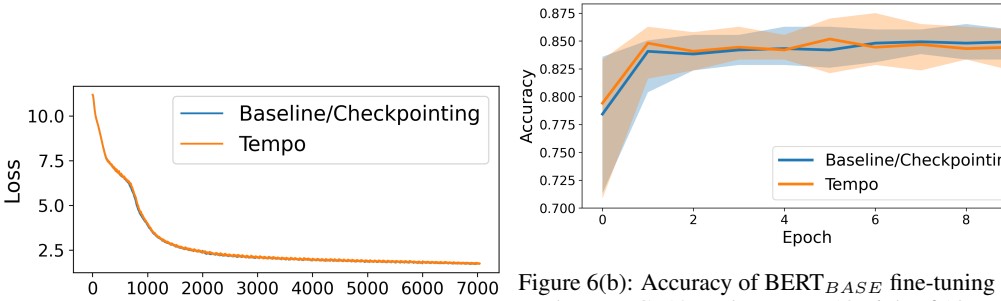

Figure 6(a): Phase 1 $\text{BERT}_{BASE}$ pre-training curve on the English Wikipedia dataset [67].

Figure 6(b): Accuracy of $\text{BERT}_{BASE}$ fine-tuning [12] on the MRPC [13] task. We run 10 trials of 10 epochs. The solid line represents the median accuracy of these trials, and the maximum and minimum along the training process by the transparent curves' boundaries.

For the fine-tuning accuracy, we use the pre-trained Huggingface [69] implementation. Figure 6b shows the results of $\text{BERT}_{LARGE}$ fine-tuning [12] on the MRPC [13] task. The figure shows a consistent overlap between the maximum and minimum accuracy of *Tempo* and the baseline, so we can conclude that *Tempo* has little impact on the accuracy of the trained model.

## 4.3   Ablation Studies

**Ablation Study With Respect to Larger Model Parameters on Modern Hardware Platforms**
We also evaluate on other hardware platforms as well as model parameters. First, we use an increased hidden layer size for various configurations. These experiments are conducted on a platform with

an NVIDIA A100 GPU [43] across sequence lengths of 128 and 512. We maintain the hidden layer size $H$ to the number of attention heads $A$ ratio of 64 which is in line with prior works [63, 12]. The results are shown in Figure 7. The figure demonstrates two important generalizations of *Tempo*. First, note that even on newer and more advanced GPUs, *Tempo* continues to provide a tangible benefit. Second, across larger hidden layer sizes *Tempo* consistently demonstrates a clear improvement over the baseline (as shown in the figure, this can be as high as a **39% speedup** over *Baseline* which corresponds to a 16% speedup over *Checkpoint*). The speedup over *Checkpoint* is as high as **20%** . We conclude that *Tempo* will continue to be applicable to new hardware and larger models.

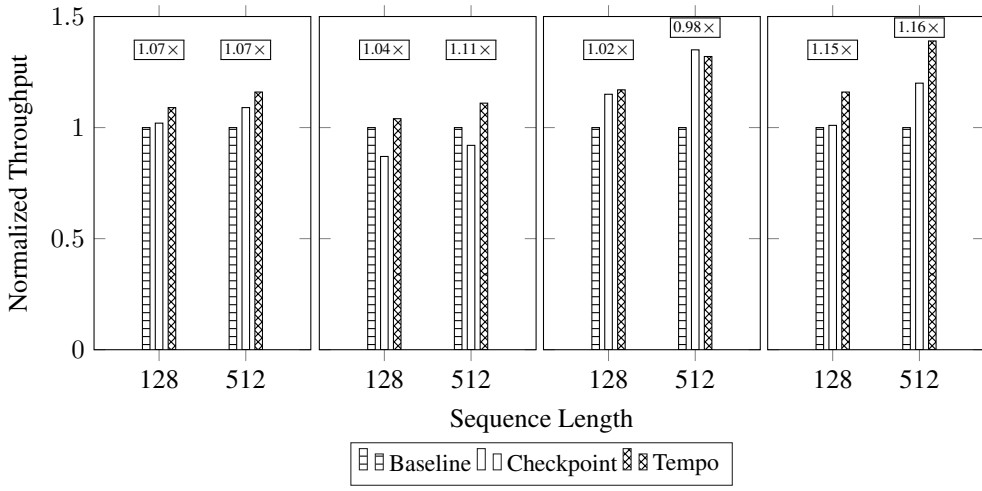

Figure 7: Normalized throughput at the maximum batch size, with annotated speedup *over the best baseline*. From left to right the configurations are (a) $\text{BERT}_{LARGE}$ ($H = 1024$), (b) $\text{BERT}_{BASE}$ $H = 2048$, (c) $\text{BERT}_{LARGE}$ $H = 2048$, (d) $\text{BERT}_{BASE}$ $H = 3072$.

We also conduct experiments on $\text{BERT}_{LARGE}$ (modified to use 12 Layers instead of 24 for more data points) for sequence lengths larger than 512. Figure 8 shows the results for this experiment, where we demonstrate that *Tempo* outperforms *Baseline* on longer sequence lengths as well which can be as high as a **27% speedup** over *Baseline*. At the same settings, we observe 16% speedup over *Checkpoint*. Tempo also outperforms *Checkpoint* by as much as **20%**. We conclude that yet again *Tempo* will be able to take advantage of modern hardware, as well as remain advantageous as sequence lengths increase. Note that the largest sequence length of 3072 on *Baseline* does not have enough memory to run.

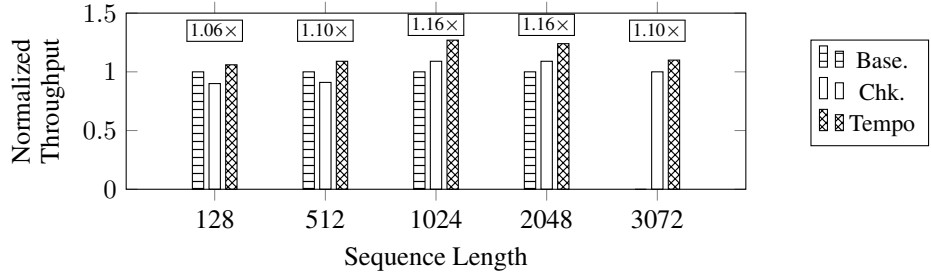

Figure 8: Normalized throughput relative to the Baseline across different sequence lengths on the NVIDIA A100 GPU for $\text{BERT}_{LARGE}$ modified to use 12 layers. We annotate each bar group with *Tempo's* speedup *over the best baseline*.

**Results on Other Models**    We conduct experiments on other Transformer-based models as well: RoBERTa [34] and GPT2 [49]. For the evaluation of RoBERTa, we use the Fairseq library [44] as well as a sequence length of 512, while GPT2 uses the Huggingface GPT2 model [69]. These experiments are conducted on both 2080 Ti and V100 setups. We note that the improvement over the baseline is substantial (up to **19% and 26%** for GPT2 and RoBERTa respectively on the 2080 Ti

setup. This improvement corresponds to a increase in batch size of $\mathbf{3\times}$ and $\mathbf{2\times}$. Furthermore, we also see speedups of 5% and 4% on the V100 setup as well. From these results, we conclude that *Tempo* generalizes well to other Transformer-based models besides BERT.

# 5    Extensions

## 5.1    Extending In-place GELU

The ideas used in section 3 for In-place GELU can be extended to general elementwise layers. The generic steps required for this are listed below, from the the high-level mathematics to the low-level kernel based accelerator implementation. To the best of our knowledge, ours is the first work that exposes this potential optimization. This is a generic strategy to reduce memory footprint in a multi-dimensional space.

Consider an elementwise layer with $n$ inputs that applies a function $f$ inputs such that $y = f(x_1, x_2, ..., x_n)$ to each corresponding element of the input tensors and where the output is retained for the backward pass of the subsequent layer.

•    Discard activation $x_1$ without loss of generality. Determine a function $g$ such that $x_1 = g(y, x_2, ..., x_n)$. For bijective functions of one variable this is simply the inverse.
•    If such a function does not exist without ambiguity, construct functions $g_1, ..., g_j$ that can recover $x_1$ on an interval. Construct a function $g_*$ such that $x_1 = g(m, y, x_2, ...x_n)$ where $m$ is an indicator that denotes the interval from which $x_1$ from and thus the piecewise selection of one of $g_1, ..., g_j$. Polynomially approximate each of $g_1, ..., g_j$ to construct a new piecewise function $g_{*p}$ in the case that they cannot be expressed analytically.
•    For the implementation of the forward pass, fold the computation of $m$ into the computation of $f$. In essence, construct a new function $f_*$ such that $(y, m) = f_*(x_1, ..., x_n)$. This can be done in a single kernel call.
•    For the backward pass, fold the calculation of $x_1 = g_{*p}(m, y, x_2, ..., x_n)$ into the computations of $\frac{\partial f}{\partial x_2}, ..., \frac{\partial f}{\partial x_n}$ if the computation of these values requires $x_1$ by composing these functions. In essence, fusing the kernels for the inverse and gradient operator. Then, we require $n$ kernel calls to calculate the gradient with respect to the loss as before.

We illustrate this strategy in appendix E in more detail. The crux of the idea is that $m$, if needed at all, can be stored with less memory than $x_1$, while keeping the number of kernel calls to a minimum.

## 5.2    Auto-Tempo

As part of exploratory future work, we consider the application of *Tempo* as an automatic compiler pass. We propose and prototype two different methods of automatically applying *Tempo* to transformers which are available at the link in section 6. The first method is a fast method of profiling beforehand to determine whether memory footprint reduction would help, then applying *Tempo* to all applicable layers. The second method is a fine-grained method applies *Tempo* to a subset of the applicable layers where the subset is determined through automatic profile and search, analogous to binary search.

# 6    Conclusion

We propose *Tempo*, a new mechanism that reduces the memory footprint of Transformer-based models at low cost. It shows an improvement in throughput of up to 16% over the state-of-the-art baseline for BERT$_{LARGE}$ pre-training task and also shows an improvement in maximum batch size of up to $2\times$ on both V100 and 2080 Ti GPUs. Our technique also generalizes well to new models, more modern hardware, as well as diverse model parameters in terms of memory footprint and throughput, demonstrating the robustness of our technique. Our hope is that *Tempo* will be used with other footprint reduction methods to improve training efficiency of Transformer-based models. We open-source *Tempo* for an immediate positive impact on both machine learning researchers and practitioners here: `https://github.com/UofT-EcoSystem/Tempo`.

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
