## Summary of Appendices

These appendices contain additional details to cover different aspects of our work: motivation, related work, implementation, methodology, and extra ablation studies.

Appendix A contains a more detailed breakdown of the memory usage of BERT$_{BASE}$.

Appendix B goes over relevant related works and the difference between them and our work.

Appendix C goes more in depth into the major memory reduction footprint techniques, and compare them to our method.

Appendix D covers details of the LayerNorm backward pass derivation.

Appendix E includes implementation details of our optimizations.

Appendix F includes additional information on the implementation of our optimizations.

Appendix G includes a more detailed description of our experimental setups.

Appendix H includes a memory footprint ablation study with respect to Tempo optimizations.

Appendix I is the NeurIPS paper checklist.

## A   A Closer Look at the Memory Breakdown of BERT$_{BASE}$

Figure 9 shows a detailed memory breakdown of the Huggingface BERT$_{BASE}$ implementation [69] on the MRPC [13] fine-tuning task at a batch size of 32, profiled using the skyline tool [70]. From the figure, encoder layer activations are clearly the major contributor to the memory footprint compared to parameter weights, gradients, and optimizer states.

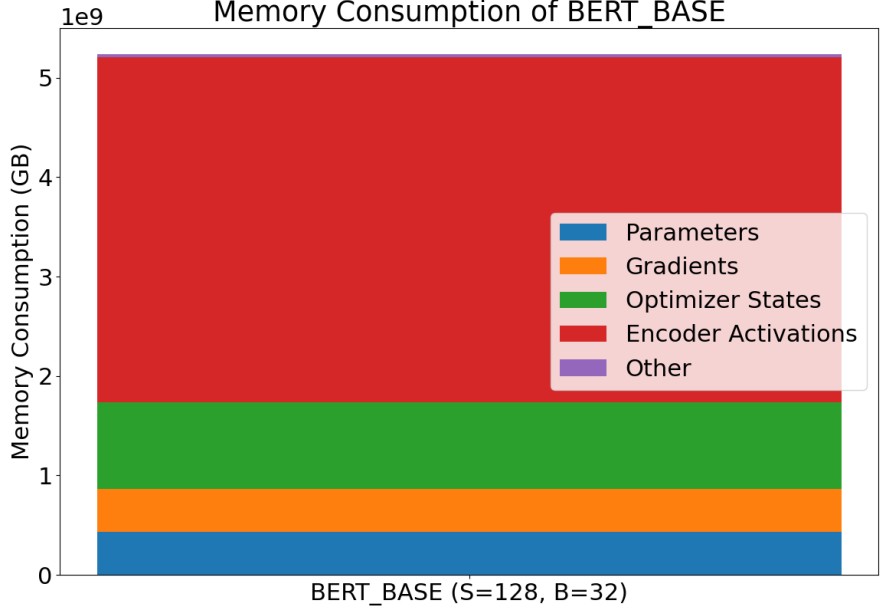

Figure 9: GPU memory breakdown for BERT$_{BASE}$ [12] fine-tuning on the MRPC [13] task using the Huggingface [69] at a sequence length of 128 and batch size of 32.

# B   Related Works

There has been a number of previous works [55, 29, 5, 11, 71] which focus on developing models that remain competitive with BERT and other Transformer-based models while requiring less memory and compute. One approach that has been taken is to tackle the $\mathcal{O}(S^2)$ nature of the attention mechanism [61]. Out of these, a few interesting and relevant ideas in terms of memory footprint reduction involve sparsifying the attention mechanism [5, 71], or using a decomposable softmax layer in order to avoid doing the $\mathcal{O}(S^2)$ computation in the first place [11]. In contrast, our technique targets a broad spectrum of sequence lengths. Other models such as TinyBERT [29] and DistilBERT [55] take a different approach in reducing the total model size. In these works, the technique of *Distillation* is used to train a smaller student network from a larger teacher network [22]. These approaches are model level algorithm changes, and therefore orthogonal to our work.

Two other important techniques are mixed precision training [36] and in-place activated Batch-Norm [53]. Mixed precision training involves training using both 32-bit and 16-bit IEEE floating point numbers depending on the numerical sensitivity of different layers [36]. This both decreases the training time and the memory pressure of the model dependent on the hardware [36]. Reversible BatchNorm uses techniques designed to optimize the memory footprint of CNN models which make extensive use of RELU and BatchNorm operators [53]. They do this by deriving in-place expressions for these operators (the input feature map for these operators no longer needs to be stored). Although this work shows better performance than checkpointing, it is also specific to CNN models, so is not directly comparable to our work. Other recent works such as Substation [24] aims to improve training performance by reducing the total movement of data with operator fusion; this is also orthogonal to our memory footprint based approach.

Other techniques are more focused on inference and model weights, both of which are not applicable in this context of Transformer-based model *training*, where the *activations* are the memory bottleneck [20, 19, 10, 25, 2, 9, 31, 45, 64].

# C   General Memory Footprint Reduction Techniques

We expand on the techniques we referred to in section 2 of checkpointing, offloading, and compression in this section. As previously iterated in section 2, (1) these techniques do not look closely at the specific structure of the BERT model, (2) At a per layer level, our techniques can provide better performance at a cheaper computational cost, and (3) these techniques are orthogonal to our work.

## C.1   Checkpointing

Early work in this direction was able to reduce the memory cost of linear models to $\mathcal{O}(\sqrt{n})$ for general linear models in the number of layers. Further work has expanded on this. Echo has innovative optimizations for specific models [73], Checkmate computes an optimal checkpointing schedule minimizing recomputation time with respect to a specific memory budget [28], and Dynamic Tensor Rematerialization uses online heuristics to minimize recomputation time while training [33]. Furthermore, there are works in this area that combine checkpointing with other techniques such as offloading detailed below.

## C.2   Offload

Initial work in this direction such as vDNN focused on specific schemes to offload layer outputs depending on their computation cost [52]. There was also some consideration into pre-fetching feature maps in anticipation of their use in the backward pass. More work in that direction focuses on several different directions. For one, Superneurons [65] combines both checkpointing and offloading. In this work, they checkpoint computationally cheap layers, avoiding transfer overhead, while simultaneously offloading computationally expensive layers, avoiding recomputation overhead for those cases. Capuchin takes this further in considering tensor level accesses, as well as considering the runtime fetching and computation time in making decisions on which strategy to apply [48]. Additionally, ZeRO-Infinity combines offloading of feature maps with offloading of model states and other optimizations [50].

## C.3 Compression

Works like Gist [26] and ActNN [6] both include forms of lossy compression. Gist specifically targets CNN based models with a variety of different lossy and lossless optimizations, taking an approach similar to our own in examining the model structure closely [26]. However, as this is a work focused on CNNs, the techniques described do not directly apply to the BERT model we are aiming to optimize for. ActNN on the other hand is a more general technique, using a quantization strategy designed to reduce the number of bits needed for feature map storage, while preserving certain theoretical guarantees regarding model conversion [6]. It additionally shows good performance relative to techniques such as Dynamic Tensor Rematerialization [33] and Capuchin [48]. This work does not consider the BERT model however [6].

# D  Backward pass of In-place LayerNorm

In the following, we show how to compute the gradients of LayerNorm layer using the output.

## D.1  Notations

We use $x$, $\hat{x}$, $y$, $\mu$ and $\sigma^2$ to represent the input, intermediate normalized input, output, and mean and variance of the input, respectively. The parameters of LayerNorm function, scaling factor and bias, are denoted by $\gamma$ and $\beta$, respectively. $L$ represents the loss. For simplicity but without the loss of generality, we assume the size of input is $(N, M)$, where the second dimension represents all the dimensions that are needed to be normalized. The meanings, definitions and sizes of variables are listed in Table 3.

| Meaning | Definition | Size |
|---|---|---|
| input | $x = \{x_{ij}, i = 1, ..., N, \ j = 1, ..., M\}$ | $(N, M)$ |
| norm-input | $\hat{x} = \{\hat{x}_{ij}, i = 1, ..., N, \ j = 1, ..., M\}$ | $(N, M)$ |
| output | $y = \{y_{ij}, i = 1, ..., N, \ j = 1, ..., M\}$ | $(N, M)$ |
| mean | $\mu = \{\mu_i, i = 1, ..., N\}$ | $(N)$ |
| variance | $\sigma^2 = \{\sigma_i^2, i = 1, ..., N\}$ | $(N)$ |
| weight | $\gamma = \{\gamma_j, j = 1, ..., M\}$ | $(M)$ |
| bias | $\beta = \{\beta_j, j = 1, ..., M\}$ | $(M)$ |

Table 3: The meanings, definitions and sizes of variables used in LayerNorm layer.

## D.2  Forward Pass

In the forward pass, input is firstly normalized along the second dimension, and then scaled and shifted accordingly.

$$\hat{x}_{ij} = \frac{x_{ij} - \mu_i}{\sqrt{\sigma_i^2 + \epsilon}}$$

$$y_{ij} = \gamma_j \hat{x}_{ij} + \beta_j$$

where $\mu_i = \frac{1}{M} \sum_{j=1}^{M} x_{ij}$ and $\sigma_i^2 = \frac{1}{M} \sum_{j=1}^{M} (x_{ij} - \mu_i)^2$. $\epsilon$ is added for numerical stability.

## D.3  Backward Pass

Our goal is to use output to compute the gradients with minimum overhead. Intuitively however, the input is needed to compute the gradients of LayerNorm, which means we need to compute backwards to get input. We find that we can use the intermediate normalized input to get what we want. The gradient derivations are listed as follows, along the lines of the BatchNorm derivation in [53].

$$\frac{\partial y_{ij}}{\partial \gamma_j} = \hat{x}_{ij}, \quad \frac{\partial y_{ij}}{\partial \beta_j} = 1, \quad \frac{\partial y_{ij}}{\partial \hat{x}_{ij}} = \gamma_j,$$

$$\frac{\partial L}{\partial \gamma_j} = \sum_{i=1}^{N} \frac{\partial L}{\partial y_{ij}} \frac{\partial y_{ij}}{\partial \gamma_j} = \sum_{i=1}^{N} \frac{\partial L}{\partial y_{ij}} \hat{x}_{ij},$$

$$\frac{\partial L}{\partial \beta_j} = \sum_{i=1}^{N} \frac{\partial L}{\partial y_{ij}} \frac{\partial y_{ij}}{\partial \beta_j} = \sum_{i=1}^{N} \frac{\partial L}{\partial y_{ij}},$$

$$\frac{\partial L}{\partial \hat{x}_{ij}} = \frac{\partial L}{\partial y_{ij}} \frac{\partial y_{ij}}{\partial \hat{x}_{ij}} = \frac{\partial L}{\partial y_{ij}} \gamma_j,$$

Here we can the gradients with regard to $\gamma$, $\beta$ and $\hat{x}$. We still need to derive the gradient to input further.

$$\frac{\partial \hat{x}_{ij}}{\partial \sigma_i^2} = -\frac{\hat{x}_{ij}}{2(\sigma_i^2 + \epsilon)}, \quad \frac{\partial \hat{x}_{ij}}{\partial \mu_i^2} = -\frac{1}{\sqrt{\sigma_i^2 + \epsilon}},$$

$$\frac{\partial L}{\partial \sigma_i^2} = \sum_{p=1}^{N} \sum_{q=1}^{M} \frac{\partial L}{\partial \hat{x}_{pq}} \frac{\partial \hat{x}_{pq}}{\partial \sigma_i^2}$$

$$= \sum_{q=1}^{M} \frac{\partial L}{\partial \hat{x}_{iq}} \frac{\partial \hat{x}_{iq}}{\partial \sigma_i^2} \quad (p = i)$$

$$= \sum_{j=1}^{M} \frac{\partial L}{\partial \hat{x}_{ij}} \frac{\partial \hat{x}_{ij}}{\partial \sigma_i^2} \quad (let \ q = j)$$

$$= \sum_{j=1}^{M} \frac{\partial L}{\partial y_{ij}} \gamma_j \cdot \left( -\frac{\hat{x}_{ij}}{2(\sigma_i^2 + \epsilon)} \right),$$

$$\frac{\partial L}{\partial \mu_i} = \sum_{p=1}^{N} \sum_{q=1}^{M} \frac{\partial L}{\partial \hat{x}_{pq}} \frac{\partial \hat{x}_{pq}}{\partial \mu_i}$$

$$= \sum_{q=1}^{M} \frac{\partial L}{\partial \hat{x}_{iq}} \frac{\partial \hat{x}_{iq}}{\partial \mu_i} \quad (p = i)$$

$$= \sum_{j=1}^{M} \frac{\partial L}{\partial \hat{x}_{ij}} \frac{\partial \hat{x}_{ij}}{\partial \mu_i} \quad (let \ q = j)$$

$$= \sum_{j=1}^{M} \frac{\partial L}{\partial y_{ij}} \gamma_j \cdot \left( -\frac{1}{\sqrt{\sigma_i^2 + \epsilon}} \right),$$

$$\frac{\partial \sigma_i^2}{\partial x_{ij}} = \frac{2(x_{ij} - \mu_i)}{M}, \quad \frac{\partial \mu_i}{\partial x_{ij}} = \frac{1}{M}, \quad \frac{\partial \hat{x}_{ij}}{\partial x_{ij}} = \frac{1}{\sqrt{\sigma_i^2 + \epsilon}},$$

Combining all the intermediate results above, we have

$$\frac{\partial L}{\partial x_{ij}} = \sum_{p=1}^{N} \sum_{q=1}^{M} \left( \frac{\partial L}{\partial \hat{x}_{pq}} \frac{\partial \hat{x}_{pq}}{\partial x_{ij}} \right) + \sum_{p=1}^{N} \left( \frac{\partial L}{\partial \sigma_p^2} \frac{\partial \sigma_p^2}{\partial x_{ij}} + \frac{\partial L}{\partial \mu_p} \frac{\partial \mu_p}{\partial x_{ij}} \right)$$

$$= \frac{\partial L}{\partial \hat{x}_{ij}} \frac{\partial \hat{x}_{ij}}{\partial x_{ij}} + \frac{\partial L}{\partial \sigma_i^2} \frac{\partial \sigma_i^2}{\partial x_{ij}} + \frac{\partial L}{\partial \mu_i} \frac{\partial \mu_i}{\partial x_{ij}}$$

$$= \left[ \frac{\partial L}{\partial y_{ij}} \gamma_j - \left( \sum_{j=1}^{m} \frac{\partial L}{\partial y_{ij}} \gamma_j \cdot \hat{x}_{ij} \right) \cdot \frac{\hat{x}_{ij}}{m} \right.$$

$$\left. - \left( \sum_{j=1}^{m} \frac{\partial L}{\partial y_{ij}} \gamma_j \right) \cdot \frac{1}{m} \right] \cdot \frac{1}{\sqrt{\sigma_i^2 + \epsilon}}$$

where intermediate normalized input $\hat{x}$ can be computed as $\hat{x}_{ij} = (y_{ij} - \beta_j)/\gamma_j$. Therefore, by extra stashing weight (scaling factor) $\gamma$, bias $\beta$ and variance of the input $\gamma^2$, we can get the gradients using output without recovering input.

# E Implementation Details

## E.1 In-Place GELU

As we show in Section 3, it is possible to compute the inverse of the GELU function by knowing what side of the minimum the input originated from. This is shown in Equation 1 where GELU* is a function that returns both the GELU output and $m$ is the mask bit that denotes which side of the minimum the input originates from.

$$\text{GELU*}^{-1}(\text{GELU}(x), m) = x \tag{1}$$

Moreover, we observe that we do not need to compute the inverse and then compute the derivative with respect to the input in a two step process as a naïve approach would suggest. Instead, we can precompute this composition (see Equation (2)):

$$\frac{d\,\text{GELU}}{dx}(y) = \text{GELU}' \circ \text{GELU*}^{-1}(y, m) \tag{2}$$

in order to **compute the derivative with respect to the input** directly using the output value. A plot of this relation is shown in Figure 10a.

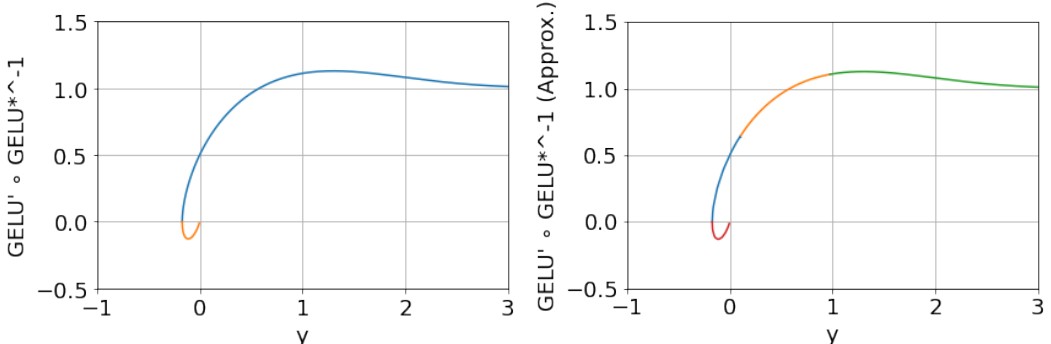

(a) GELU derivative from (2). The section corresponding to $x > -0.75179$ is in blue, $x \leq -0.75179$ – in orange.

(b) Our approximation of function (2) with a piece-wise polynomial approximation. Different sections of the approximation are shown in different colours.

A key observation about the GELU function is that it is transcendental, and hence there is no simple solution for the inverse of the GELU function in terms of elementary functions [58]. We therefore approximate sections of Equation (2) with a piece-wise polynomial with a degree up to 13.[5] A plot of this approximation is shown in Figure 10b.

## E.2 In-Place LayerNorm

As stated in Section 3, we aim to reuse the output of the LayerNorm [4], which must be stored for the backpropagation of the successive fully-connected layers, while discarding the input. Although prior work has demonstrated the usability of In-Place Activated BatchNorm in the context of CNN networks [53], we note that this approach is not applicable in the Transformer case, which employs LayerNorm instead [63].

By employing an alternative derivation for the gradient of LayerNorm which stashes alternative parameters, we can compute the gradients with *negligible* performance overhead while achieving ideal memory footprint reduction for this operator. Following a similar approach as for In-Place GELU, we implement this operator as a Python PyTorch module, allowing it to be easily substituted in place of the existing LayerNorm layer in an implementation of the BERT model [41, 69]. See F for additional implementation details and the full derivation.

---

[5]Additional details of this implementation are covered in Appendix F.

### E.3 Sub-Layer Dropout Recomputation

Our basic implementation of Sub-Layer Dropout Recomputation follows the example of our other optimizations. Note that the way Dropout is implemented requires a randomly generated mask, where a portion of the inputs are set to zero according to a percentage $p$, which is also needed in the backward pass [46, 7]. We simply stash this mask, and discard the output. Then, in the backward pass, we recompute the output as shown in Figure 4. Storing a boolean mask of size $N$ will take $N \times 1$ bytes, boolean tensors in PyTorch use 1 byte per value, while 32-bit floating points will use $N \times 4$ bytes [46]. Therefore, the total memory saved by discarding the output will be $4/5$ of the total pre-optimization dropout output memory cost.

In contrast to prior works which modify the framework and may not expose this optimization [48, 32, 27], we develop a PyTorch module which can be added in to reduce the memory pressure of the critical attention section of the Transformer-based models [63] with minimal overhead. Appendix F provides more in-depth comparison between prior work and our implementation.

### E.4 Other Engineering Optimizations

We note that PyTorch uses a memory-inefficient implementation of the softmax function which retains both the input and output of the function for the backward pass [46]. Instead, only the output is necessary. This optimization has also previously been implemented as part of some models in the Huggingface library [18]. We use this optimization as well in our implementation of the attention mechanism.

### E.5 Elementwise Extension

We illustrate the difference between our general elementwise strategy in Figure 11.

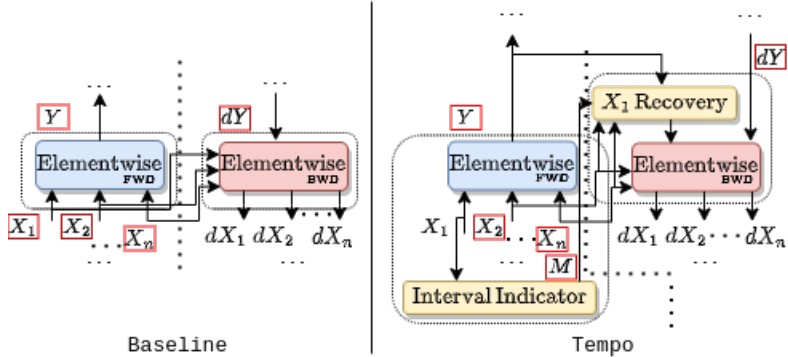

Figure 11: Comparison of our in-place general elementwise strategy with the baseline. Dotted border rectangles enclose functions that can be executed in a single kernel. Red bordered activations and gradients are needed for the computations of the elementwise and successive layer's backward pass.

## F Additional Implementation Details

This section contains additional implementation details of our technique.

### F.1 In-place GELU

We implement this optimization in PyTorch. In this case, we write the forward pass to return the GELU of the input function, as well as a boolean mask indicating whether the input is greater than or equal to the point at which the minimum value occurs, $x \approx -0.75179$. The backward pass is slightly more complicated. We take as inputs to the backward pass the incoming gradient, the saved mask, as

well as the saved output values. The corresponding approximating polynomial shown in Figure 10b is determined and then computed. We use approximating polynomials of up to degree 13 in this case.

We implement the forward and backward pass using CUDA [37] which is wrapped in C++ [23]. Both of these are wrapped in a Python [62] PyTorch layer to be substituted easily for an existing implementation. While profiling our implementation we found that the memory latency of loading these inputs, as well as storing the output to be the bottleneck. We were thereby able to implement polynomials of degree 13, since the computation is hidden by the memory access latency, although better approximations may be possible.

### F.2 In-place LayerNorm

In-place Layernorm is implemented as a custom PyTorch module [46]. This is done in 3 stages as per PyTorch's custom module implementation. To do this, we write a custom CUDA [37] implementation based on PyTorch's own implementation of the LayerNorm layer [46]. We write a forward and backward layer wrapper in C++ [23], which is then again wrapped as a Python [62] PyTorch module, allowing it to be easily substituted in place of the existing LayerNorm layer in an implementation of the BERT model such as the NVIDIA DeepLearningExamples BERT implementation or the Huggingface implementation [41, 69].

### F.3 Sub-Layer Dropout Recomputation

We note that although the state-of-the-art recomputation/checkpointing papers [28, 33] consider such an abstract idea in theory, their practical implementations [27, 32] never treat the sub-layer granularity provided in the frameworks as the lowest granularity of those techniques' applicability. We concede that in the case of TensorFlow, the dropout layer has additional underlying granularity as a result of its implementation [1], and hence this would not be applicable in that case – and Checkmate [28] would consider this level of granularity. However, this is not the case in PyTorch's checkpointing implementation [16]. The tensor-level granularity of optimization is noted in Capuchin [48] as well. However, Capuchin is a technique that requires runtime level profiling, and modifications to the framework. It may or may not offload, recompute, or otherwise store any part of the dropout layer specified.

Our method uses in-built PyTorch operators at a C++ level to rewrite the attention mechanism, which is again wrapped as a Python PyTorch module to be substituted.

## G  Experimental Setup

Table 4 shows a more detailed view of our experimental platform.

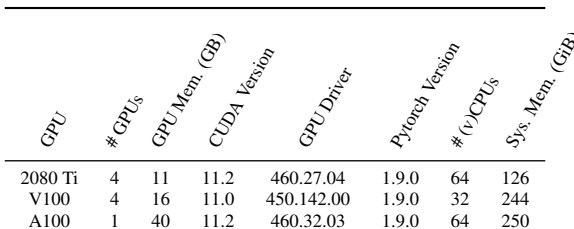

| GPU | #GPUs | GPU Mem. (GB) | CUDA Version | GPU Driver | Pytorch Version | #(v)CPUs | Sys. Mem. (GiB) |
|---|---|---|---|---|---|---|---|
| 2080 Ti | 4 | 11 | 11.2 | 460.27.04 | 1.9.0 | 64 | 126 |
| V100 | 4 | 16 | 11.0 | 450.142.00 | 1.9.0 | 32 | 244 |
| A100 | 1 | 40 | 11.2 | 460.32.03 | 1.9.0 | 64 | 250 |

Table 4: A short summary of our test setups, including CUDA [37], PyTorch [46], and driver [38] versions.

## H  Memory Footprint Reduction Ablation Study

We calculate the memory footprint reduction contributed by each optimization across different sequence lengths relative to the total memory footprint of each encoder layer, given a $H$ to $A$ ratio of 64 [63, 12]. This is shown in Figure 12 for the selected configurations. From the figure, it's clear that In-Place GELU and LayerNorm provide the bulk of the memory footprint reduction in the

short sequence length regime, while the other two optimizations provide an improvement in the long sequence length case. Note that this is due to the fact that the latter's memory footprint reduction is $\mathcal{O}(S^2)$, while the former's memory footprint reduction goes as $\mathcal{O}(SH)$ where $S$ is the sequence length. This allows *Tempo* to stay robust to model parameters, and provide consistent performance across sequence lengths.

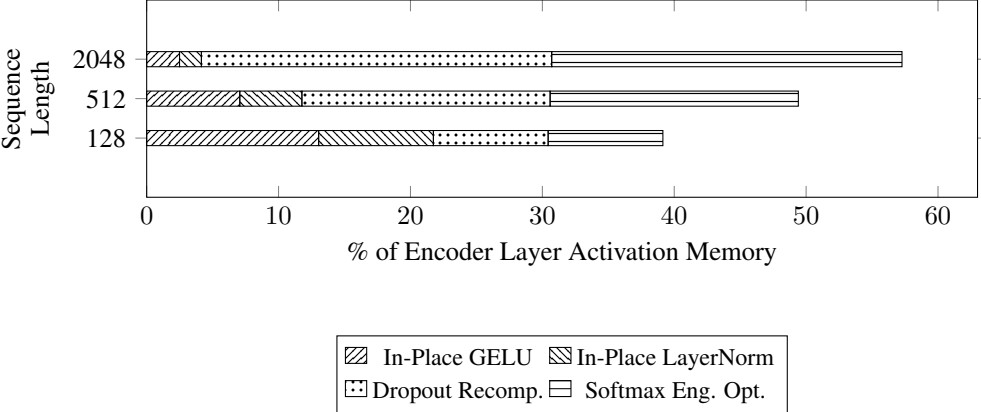

Figure 12: Per layer comparison of Tempo memory footprint reduction across different sequence lengths.