# OpenReview forum: "Tempo: Accelerating Transformer-Based Model Training through Memory Footprint Reduction"
_NeurIPS.cc/2022/Conference — NeurIPS 2022 Accept_

### Official Review · Reviewer_mcUf · 2022-06-25

**Rating:** 6
**Confidence:** 4
**Soundness:** 3 good
**Presentation:** 2 fair
**Contribution:** 3 good

**Summary:**

The paper proposes optimizations that reduces memory usage when training transformer models. Reducing memory usage for training models allows using larger batch size which improves training throughput.
Previous methods that reduce memory footprint in training are: checkpointing, offloading and compression.
The changes presented in the paper are: in-place GELU layer, in-place Layer Normalization and Dropout.
The proposed method was evaluated on BERT, RoBERTa, GPT2 model. They show improved training throughput and ability to use larger batch sizes when training BERT with their method.




**Questions:**

Some suggestions and questions that could improve the paper are:

Adding a comparison to check checkpoint+tempo to demonstrate that tempo can be used with other techniques

Adding a comparison with compression techniques to demonstrate the trade-off of accuracy loss with more efficient training from tempo

Try on more models that uses GELU to demonstrate that this method isn't only suited for transformers

Having a list of layers from various models (CNN, RNN, GANs, etc) that could be done in-place (that arent done yet) to improve memory footprint in training would be a value study

Table2 should also list and highlight throughput achieved and have total memory usage

Figure3 caption format seems confusing




**Limitations:**

The paper doesn't discuss limitations or future work.
Discussion how improving training on transformer models can lead to lower power consumption in servers systems.
And add future work on what other layers could be done in place.

**Strengths And Weaknesses:**

Pros:
The paper demonstrates techniques to improve training of relevant transformer models using in-place layer compute. The paper tackles an important problem

Overall the paper is clear. But improvements on diagrams could improve clarity of the paper

Cons:
It is dubious that this is the first paper to analyze transformer training memory footprint

in-place compute is fairly known optimization method for training thus the method lacks originality

More experiments and comparison could be added to strengthen the paper

---

> ### Author Response · Authors · 2022-08-02
> **Rebuttal**
>
> We thank the reviewer for the valuable feedback. We address the reviewer’s questions and concerns below.
>
> 1. Novelty of Applicability to Transformer
>
> We address this in our motivation. Ours is, to the best of our knowledge, the first to analyze the Transformer model memory footprint at this granularity. The literature looking at a similar level of granularity is sparse, with the works we know of applying to CNNs ([26, 53]). Usually more studied forms of memory footprint reduction, which happen to be general, are used, and so miss out on the opportunities we show [8, 52, 6]. Furthermore, note that MFR is not often the first thing considered for throughput improvement anyways. Please kindly provide a reference if we missed any key prior works.
>
> 2. Novelty of Techniques
>
> While it is true that In-place operations are supported, for example in PyTorch, this is mostly for inference. From PyTorch documentation: “Supporting in-place operations in autograd is a hard matter, and we discourage their use in most cases.” [R1]. When it comes to training note the two works above [26, 53] are the ones we are familiar with, which again deal with different layers applicable to CNNs.
> In anticipation of the question on how our techniques add value and novelty beyond being applicable to Transformers, we list the insights required for GELU and compare it to ReLU, which we believe on its own is an interesting research contribution.
>
> 1. We want to calculate the gradient using the output feature map => must invert (Not the case with ReLU, where we can calculate the gradient using the same derivative operator on the output that we would use on the input; we just set non-zero values to 1).
>
> 2. But we cannot invert, GELU is not bijective => store a boolean to allow invertibility in conjunction with the stored output. (Contrast this to ReLU where the 1-bit stored tensor is stored for a fundamentally different reason. In the ReLU case the 1-bit tensor is the derivative itself, which is computed in advance. It is not even necessary to store the output, since the derivative is already calculated [26].)
>
> 3. No closed form for the inverse of bijective segments => must approximate. (No analogue exists for ReLU)
>
> 4. Not better than checkpointing the input, then recalculating the output during the backwards pass in terms of runtime => fuse the derivative operator kernel with the inverse approximation kernel in one composite derivative-from-output kernel. (No analogue exists for ReLU)
>
> To the best of our knowledge no previous work has shown any similar way to deal with general activation functions. This itself is a novel contribution that would be of interest to the wider community.
>
>
> 3. Checkpointing + Tempo
>
> While this is interesting, this is outside the scope of our work which is to focus on our fine-grained techniques. Not only does this require significant engineering effort, but also space in the paper which we felt was better allocated to exposition of our technique.
>
> 4. Comparison with Compression
>
> The exact approximation used for our GELU technique can be made arbitrarily accurate. This is in contrast  to compression-based techniques, where the fact that error is introduced is fundamental to the technique. We can always find better approximating polynomials, but compression is “stepwise” so there is no such guarantee here. We empirically find that four degree 13 polynomials applied piecewise are still able to achieve good results as shown in the evaluation.
> But even more accurate polynomial approximations are still practical. For example, a careful application of Horner’s method means that each polynomial term requires only one FMA operation, and $\mathcal{O}(log(n))$ conditional branches where $n$ scales as the number of pieces in the approximation. When we combine these cheap operations with the fact that with elementwise kernels the memory access time is high latency, our back of the hand calculations find that $200$ terms is more than feasible with minimal effect on the computation time.
>
> 5. Additional Layer Types
>
> As shown above, our GELU technique can be easily generalized to other activation functions. However, this is again outside the scope of our work which is focused on the Transformer-based models, and would be hard to justify over more exposition on our technique.
>
> 6. Limitations
>
> In section 1, we briefly touch on the positive impact of our proposal related to cost savings and carbon footprint reduction [24, 60] - our technique can decrease the total training time and thus reduce the energy required. We also touch on limitations of our work with respect to total memory footprint reduction and lossiness.
>
> 7. Final Thoughts
>
> Our work has original insights not present in other works, and shows a significant performance improvement on the important Transformer workload. This would make our work a good fit for NeurIPS as part of its Infrastructure focus.
>
> [R1] https://pytorch.org/docs/stable/notes/autograd.html

---

> > ### Comment · Reviewer_mcUf · 2022-08-05
> > **Rebuttal acknowledged**
> >
> > Thanks to the authors for the rebuttal
> >
> > The implementation effort and addition to infrastructure is valuable and it achieves good results on currently meaningful transformer models.
> >
> > The concern is the impact of the work. Question that appears are: can it be automated in compilers for other models (non transformer models), what are benefits from other layers, what if transformers with some other non activation layer becomes the state of the art.
> >
> > If the authors could address those it would strengthen the great paper

---

> > > ### Author Response · Authors · 2022-08-06
> > > **Generality Concerns**
> > >
> > > We thank the reviewer for the reply.
> > >
> > > 1. In fact, we have already developed this automatic technique, which will automatically apply our in-place optimisations for any PyTorch module (not just Transformers) as long as it uses one of the layer types in our paper by replacing the layer with its equivalent memory-efficient version. Furthermore, our automatic technique determines whether our memory optimisations are worth applying by profiling the model beforehand using one of two methods, depending on whether a fast optimization or a more thorough optimization is desired.
> > >
> > > - Fast Method: Profile the unoptimized model until the highest batch size for a given input size. Use the curve of forward+backward time versus batch size to determine whether the model is in one of the following regimes and then optimize the layers automatically based on this:
> > >     - Saturated regime: where increasing the batch size does not improve the throughput significantly (and thus unlikely to benefit from more efficient memory use)
> > >     - Unsaturated regime: where the small batch size means the compute resource of the GPU is not fully utilized (and thus likely to benefit from our memory-footprint reducing optimisations).
> > >
> > >   Prior work shows examples of models which fit these two categories, see Figure 3 from Echo [73]. This allows us to quickly and automatically return an optimized model if it will lead to better performance.
> > >
> > > - Fine-grained Method: For a more thorough method, we profile both optimized and unoptimized versions of the model to get performance numbers. Then, the technique uses binary search to apply optimisations to only half the applicable layers. The two highest performance configurations are taken and this is repeated again with the midpoint. This is iterated for a fixed number of trials, and then the best performing model is returned.
> > >
> > > 2 & 3. Since the second and third points are related we address them together. As we highlighted in our rebuttal, the steps that we listed above for GELU are applicable even if other layers are used, as long as they are elementwise.
> > >
> > > For example, as the reviewer noted there have been attempts to replace the activation in Transformer feed forward networks with alternative mechanisms such as GLU [R2]. Our method could still be applied here. The activation functions which are used as part of GLU can be dealt with based on the four insights given in the rebuttal. To quickly illustrate something different, we take the componentwise multiplication used in GLU as well, say $Y = A \circ B$ assuming we need to retain $Y$. In this case our steps apply by storing only one of $A$ or $B$ as well as $Y$. Without loss of generality we store $B$. Then, the gradients can be computed by doing $\frac{\partial \mathcal{L}}{\partial A} = \frac{\partial \mathcal{L}}{\partial Y} \circ B$ and  $\frac{\partial \mathcal{L}}{\partial B} = \frac{\partial \mathcal{L}}{\partial Y} \circ Y \circ (1/B)$. These can be done in two kernel calls, the same as the standard but with less memory used. Contrast this to checkpointing the layer which would take an extra kernel call. Prior works that deal with CNNs do not deal with this as well [26, 53]. These can all be done without steps 2 and 3 above which further serves to illustrate the generality of our steps.
> > >
> > > We will include these steps as they apply to GELU and how they can be applied in general so that others can leverage our ideas and optimize any elementwise layers, which are ubiquitous in state-of-the-art models from CNNs to RNNs [R3, R4].
> > >
> > > We thank the reviewer again for the valuable feedback and will include both points in the revision.
> > >
> > > [R2] N. Shazeer, "GLU Variants Improve Transformers", https://arxiv.org/pdf/2002.05202v1.pdf
> > > [R3] "Deep Residual Learning for Image Recognition", https://arxiv.org/pdf/1512.03385.pdf
> > > [R4] "Learning Phrase Representations using RNN Encoder–Decoder for Statistical Machine Translation", https://arxiv.org/pdf/1406.1078v3.pdf

---

> > > > ### Comment · Reviewer_mcUf · 2022-08-09
> > > > **Rebuttal acknowledged #2**
> > > >
> > > > Thanks to the authors for proving more info and additional data in the paper

---

### Official Review · Reviewer_Qdmx · 2022-07-10

**Rating:** 6
**Confidence:** 4
**Soundness:** 3 good
**Presentation:** 3 good
**Contribution:** 3 good

**Summary:**

This paper proposes an approach to reduce the memory footprint of transformer-based models for faster training without sacrificing accuracy. The proposed approach Tempo uses In-Place GELU, In-Place LayerNorm, Sub-Layer Dropout Re-computation, and softmax engineering optimization. The main advantage is to compute the backward pass as normal while requiring less storage. Evaluation results show 2x improvement in batch size for BERTLARGE pre-training at a sequence length of 512 while increasing the training throughput by up to 16%.

**Questions:**

If there are any data for the above issue 3, please briefly summarize.

**Limitations:**

Yes.

**Strengths And Weaknesses:**

Strengths:
+ Clever techniques to enable recomputing input from output in GELU and LayerNorm with minimal overhead
+ Good observation on dropout that avoids recomputing of unnecessary input
+ Well organized paper and clear writing. Supplemental materials are helpful

Weaknesses:
1. Tempo requires more memory than checkpointing (although expected, this is still a fact/weakness)
2. Optimizations in Tempo are limited to transformer-based models while some other alternatives are not (e.g., [48][28][26])
3. Evaluation-related:
- In Section 2.1, a sequence length of 512 is used for showing that feature maps account for 56% of the encoder layer activation memory, but a sequence length of 128 is used for showing that the memory function takes 17%. It would be more convincing if both results were given for the same sequence length.
- In ablation studies, it would be great to show the impact on accuracy and memory footprint when increasing the number of hidden layers.

Minor issues and suggestions:
- Figures 3 and 6 are missing the main caption.
- GPU names can be added as a row in Table 2.

---

> ### Author Response · Authors · 2022-08-02
> **Rebuttal**
>
> We thank the reviewer for the valuable feedback. We address the reviewer’s questions and concerns below.
>
> #### 1. Weakness 1 and 2
>
> With respect to these weaknesses, we would just like to briefly reiterate how our technique is orthogonal to checkpointing, and that the results that we are able to achieve require this level of fine-grained optimization of the Transformer model which allows us to take advantage of things other techniques overlook. Furthermore, note how, as we evidence in the introduction, the ubiquity and importance of the Transformer, which makes it a crucial workload to analyze in its own right.
>
> #### 2. Weakness 3
>
> We cover this in more detail in Appendix H, especially the graph which has a more detailed breakdown. To answer the question directly though, at a sequence length of 128 the three $\mathcal{O}(S^2)$ feature maps take 26.1% while the GELU at a sequence length of 512 takes 9%. To clarify the writing, which is also further explained in Appendix H, what we hope to highlight is how different feature maps are the bottleneck at different sequence lengths. This is due to the memory of some going as $\mathcal{O}(S^2)$ and others as $\mathcal{O}(SH)$. In order for our technique to be effective across all sequence lengths, it should target both of these regimes.
>
>
> The impact on the percentage memory footprint reduction is linear with respect to the number of hidden layers for a given sequence length. We conducted the ablation study with respect to the sequence length in Appendix H.
>
>
> We would like to point out in the original BERT paper [12] the authors note that an increased number of layers gives better accuracy. We expect the trend to be the same. Furthermore, we would like to point out that the lossy approximation of the GELU inverse-derivative operator can be made arbitrarily accurate. We empirically found that four degree 13 polynomials applied piecewise are still able to achieve good results as shown in the evaluation.
>
> Moreover, continuing with polynomial approximations, much better approximations are quite practical. Two points that are useful to note here are that a careful application of Horner’s method means that each polynomial term requires only one FMA operation, and $\mathcal{O}(log(n))$ conditional branches where $n$ scales as the number of pieces in the approximation. When we note that in elementwise kernels the order of magnitude of global memory accesses is much higher than the computation time, our back of the hand calculations show that approximations of up to $200$ terms is more than feasible with minimal effect on the computation time!
>
> Contrast this to compression based techniques, where the level of lossiness and the fact that there is error is an inherent part of the technique.
>
> So, as the approximation improves, there would be less of a worry with respect to accuracy versus number of hidden layers.
>
> #### 3. Final Thoughts
>
> We would like to reiterate that our work represents a significant performance improvement that is overlooked by other works for an important workload such as the Transformer. We hope that we have convinced the reviewer that our work is of interest to ML researchers and practitioners, and therefore a good fit for NeurIPS as part of its Infrastructure focus.

---

> > ### Comment · Reviewer_Qdmx · 2022-08-09
> > **Thank you for the response**
> >
> > I would like to thank the authors for providing detailed response. I remain positive on this paper.

---

### Official Review · Reviewer_WUQt · 2022-07-11

**Rating:** 5
**Confidence:** 4
**Soundness:** 3 good
**Presentation:** 3 good
**Contribution:** 2 fair

**Summary:**

This paper proposes a solution named Tempo that includes three core techniques, in-place GELU, in-place LayerNorm and sub-layer dropout recomputation, to reduce the memory footprint of training transformer-based models. Given the reduction of memory usage, the same GPU is able to conduct gradient-based optimization with larger mini-batches; therefore, the training throughput is improved.

**Questions:**

It seems the experiments are conducted on multi-GPU machines, Is there any parallel schema applied in these benchmarks?


**Limitations:**

Not applicable.

**Strengths And Weaknesses:**

Strengths:

- The out of memory issue is a serious problem when training transformer-based models, especially giant foundation models.

- The analysis of the memory consumption of transformer models during the training procedure is informative.

- The performance boost over the state-of-the-art baseline and checkpoint approach is significant.

Weaknesses:

- The novelty inherited from the proposed methods is relatively limited. These optimizations seem straightforward from the point of view of implementation.

- There is a lack of theoretical discussion about the lossy compression in the Tempo system.

---

> ### Author Response · Authors · 2022-08-02
> **Rebuttal**
>
> We thank the reviewer for the valuable feedback. We address the reviewer’s questions and concerns below.
> #### 1. Novelty
>
> Although the implementation may seem simple, there are multiple insights involved which make it non-trivial. We demonstrate this by listing the insights required for GELU and comparing it to ReLU, which on its own is a novel contribution. These steps require looking at the function from the high-level mathematics down to low-level implementation details in order to make it a viable alternative to simply checkpointing the GELU function.
>
> 1. We want to calculate the gradient using the output feature map => must invert (Not the case with ReLU, where we can calculate the gradient using the same derivative operator on the output that we would use on the input; we just set non-zero values to 1).
> 2. But we cannot invert, GELU is not bijective => store a boolean to allow invertibility in conjunction with the stored output. (Contrast this to ReLU where the 1-bit stored tensor is stored for a fundamentally different reason. In the ReLU case the 1-bit tensor is the derivative itself, which is computed in advance. It is not even necessary to store the output, since the derivative is already calculated [26].)
> 3. No closed form for the inverse of bijective segments => must approximate. (No analogue exists for ReLU)
> 4. Not better than checkpointing the input, then recalculating the output during the backwards pass in terms of runtime => fuse the derivative operator kernel with the inverse approximation kernel in one composite derivative-from-output kernel. (No analogue exists for ReLU)
>
> To the best of our knowledge no previous work has shown any similar way to deal with general activation functions. Indeed, the literature on using similar in-place memory footprint reduction ideas itself is very sparse and limited to CNNs (Gist [26] and In-Place ABN [53]). This itself is, in our opinion, a novel contribution that would be of interest to the wider community.
>
> #### 2. Lossiness
>
> We would like to point out that compression is not used as part of our technique. Instead, the lossiness would come from step 3 in the above section, which is our approximation for the composite inverse-derivative function that we employ with GELU. With regard to this, we would like to point out that the particular approximation used, or the lossiness due to it, is not an inherent part of our technique - our contribution is that it can be done, and as demonstrated in our results with minimal impact on pre-training loss and fine-tuning accuracy. The exact approximation can be made arbitrarily better. We empirically found that four degree 13 polynomials applied piecewise are still able to achieve good results as shown in the evaluation.
> Furthermore, continuing with polynomial approximations, much better approximations are quite practical. Two points that are useful to note here are that a careful application of Horner’s method means that each polynomial term requires only one FMA operation, and $\mathcal{O}(log(n))$ conditional branches where $n$ scales as the number of pieces in the approximation. When we combine these cheap operations with the fact that in elementwise kernels the order of magnitude of global memory access latency is much higher than the computation time, our back of the hand calculations find that $200$ terms is more than feasible with minimal effect on the computation time. This is different from compression based techniques, where the level of lossiness and the fact that there is error is an inherent part of the technique due to its “stepwise” nature.
>
> #### 3. Multi-GPU
>
> The script we use combines flat_dist_call from the NVIDIA Apex library which uses torch.distributed.broadcast to broadcast parameters, so it is a data parallel scheme. The exact call is linked here: https://github.com/NVIDIA/DeepLearningExamples/blob/7a4c42501ce05ac5b76999e3b9ddadeffd177b1d/PyTorch/LanguageModeling/BERT/run_pretraining.py#L423
> Our technique is only really concerned with the forward/backward semantics on a single GPU however.
>
> #### 4. Final Thoughts
>
> We would like to reiterate that our work has original insights and novelty not present in other works as shown above and represents a significant performance improvement that is overlooked by other works for an important workload such as the Transformer. We hope that we have convinced the reviewer that our work is of interest to ML researchers and practitioners, and therefore a good fit for NeurIPS as part of its Infrastructure focus.

---

> > ### Comment · Reviewer_WUQt · 2022-08-08
> > **Thanks for the response!**
> >
> > I want to thank the author for the detailed response.
> >
> > The clarification of the lossy compression is helpful for me to understand the technique.
> >
> > The explanation of Multi-GPU setting is also helpful, I think it would be helpful to include such details in the appendix.
> >
> > I think we agreed on the statement that this technique would be helpful for efficient training of memory-intensive DNNs.

---

### Official Review · Reviewer_YdHM · 2022-07-13

**Rating:** 5
**Confidence:** 3
**Soundness:** 3 good
**Presentation:** 3 good
**Contribution:** 3 good

**Summary:**

This paper proposes three simple yet effective optimizations to speedup the pre-training of large language models by reducing the memory footprint. The resulting approach is called Tempo. Tempo is able to reduce the memory requirement for the pre-training of language models such as BERT, RoBERTa, and GPT-2 significantly. In addition, the authors demonstrate that memory savings can be translated into increased throughput.

**Questions:**

1. Could you elaborate on how in-place GELU and layer normalization are different from in-place ReLU and batch normalization? Are there any new treatments involved in converting these two operations in-place?
2. Have you evaluated the pre-training performance on other libraries with complier optimization enabled (e.g., Pytorch with JIT or Tensorflow with XLA enabled)? Is it possible that some compiler optimizations, such as operation/kernel fusion, have helped to reduce memory footprint and improve training throughput? If so, are the proposed optimizations complementary to the existing compiler optimizations? For example, Pytorch performance optimization guide [1] uses GELU as an example for kernel fusion as follows:
```
@torch.jit.script
def fused_gelu(x):
    return x * 0.5 * (1.0 + torch.erf(x / 1.41421))
```

[1] https://pytorch.org/tutorials/recipes/recipes/tuning_guide.html

**Strengths And Weaknesses:**

Strengths: The three proposed optimizations for reducing the memory footprint are simple and straightforward. I am convinced that the proposed ideas can help reducing the memory requirement for the pre-training of language models significantly. The paper is well written and easy to understand. The experimental evaluation is solid, as the authors demonstrate not only the results of reduced memory footprint, but also the improvement in throughput.

Weaknesses: My main concern with this paper is whether the proposed optimizations are considered as new contributions. In particular, the first two optimizations propose to turn the activation and normalization functions into in place functions, which helps to reduce the required memory for saving the intermediate tensor for backward propagation. The idea of using in-place function to reduce memory footprint is not new. Although not directly applicable to the Transformer architecture, some previous work and in-place ABNs have proposed to perform activation and batch normalization in-place to reduce memory footprint. In addition, attempts have also been made to add support for in situ GELU in pytorch (https://github.com/pytorch/pytorch/pull/74629).

---

> ### Author Response · Authors · 2022-08-02
> **Rebuttal**
>
> We thank the reviewer for the valuable feedback. We address the reviewer’s questions and concerns below.
>
> #### 1. Novelty of contribution.
>
> We refer the reviewer to Appendix E for a more detailed overview of the techniques themselves. Particularly with respect to GELU, the novelty of our approach comes from the fact that  ReLU-focused optimizations cover only this special case, while the technique we develop for GELU is a lot more general.
>
> Note two special facts about the ReLU function:
> * If we discard the input but retain the output, we can still calculate the gradient easily by applying the exact same derivative operator that we would have applied to the input on the output as well (step function).
> * The derivative can be stored in a one-bit format, and precomputed at the time of the forward pass allowing both input and output to be discarded [26].
>
> Neither of these nice properties are applicable to GELU, or indeed in general for that matter.
>
> There are multiple insights made by our work due to the complexity of this function from the high-level math to the low-level implementation details in order to make the GELU function in-place. Such complexity does not exist in the case of ReLU. We highlight these points explicitly here.
>
> 1. We want to calculate the gradient using the output feature map => must invert (Not the case with ReLU, where we can calculate the gradient using the same derivative operator on the output that we would use on the input; we just set non-zero values to 1).
> 2. But we cannot invert, GELU is not bijective => store a boolean to allow invertibility in conjunction with the stored output. Contrast this to ReLU where the 1-bit stored tensor is stored for a fundamentally different reason. In the ReLU case the 1-bit tensor is the derivative itself, which is computed in advance. It is not even necessary to store the output, since the derivative is already calculated [26].
> 3. No closed form for the inverse of bijective segments => must approximate. (No analogue exists for ReLU)
> 4. Not better than checkpointing the input, then recalculating the output during the backwards pass in terms of runtime => fuse the derivative operator kernel with the inverse approximation kernel in one composite derivative-from-output kernel. (No analogue exists for ReLU)
>
> To the best of our knowledge no previous work has shown any similar way to deal with general activation functions. Indeed, the literature on using similar in-place memory footprint reduction ideas itself is very sparse and limited to CNNs (Gist [26] and In-Place ABN [53]). This itself is a novel enough contribution that would be of interest to the wider community.
>
>
> #### 2. PyTorch GitHub In-Situ GELU
>
> From our cursory glance at the reviewers linked GitHub PR, (in place GELU), this recent update focuses on inference only, which is not comparable to our training focused work that requires the above insights. For example, the aforementioned invertibility challenge which is required for training is not addressed in this GitHub PR to the best of our knowledge.
>
> #### 3. Kernel Fusion
>
> We note that kernel fusion may improve training throughput by making less memory accesses as shown in the link provided by the reviewer, but this will not affect the memory footprint. For example the GELU example in the review may coalesce the multiple elementwise operations (addition, multiplication, erf) into one kernel as is the case with just using the regular pytorch GELU function (torch.nn.GELU) directly, but it does not affect the amount of feature maps required for the GELU activation. Our technique involves operators which are at a mathematical level different but are functionally equivalent to the original operators which is how we are able to reduce the memory footprint.
> Kernel fusion passes can be applied on top of our optimizations, and are therefore orthogonal, but this requires a large engineering effort and is also out of the scope of our work in the same way as mixed precision training and alternative attention mechanisms. These are discussed in appendix B.
>
> #### 4. Final thoughts
>
> We would like to reiterate that our work has original insights and novelty not present in other works as shown above and represents a significant performance improvement that is overlooked by other works for an important workload such as the Transformer. These two points would make our work of interest to ML researchers and practitioners, and therefore a good fit for NeurIPS as part of its Infrastructure focus.

---

> > ### Comment · Reviewer_YdHM · 2022-08-08
> > **Post-rebuttal**
> >
> > Thanks authors for their detailed response. The response addressed most of my concerns. I will raise my rating to borderline accept.

---

### Meta-Review · Area_Chair_QX8N · 2022-08-25

**Recommendation:** Accept
**Confidence:** Certain

**Metareview:**

The paper is well organized, the writing is clear, the proposed method shows good performance and the empirical results are convincing. The main concerns that came up in the reviews concerned the novelty of the approach because similar memory-footprint optimization techniques have been used in other contexts. The authors provided valuable clarifications in the response regarding the challenges that are particular to GELU and explained what puts their approach apart from existing uses of in-place operations.

Overall, the rebuttal clarified most of the concerns, the sentiment among the reviewers is positive and I think the contribution is sufficiently novel and targets a very important use case (transformer models). Thus, I recommend acceptance, but I would recommend the authors to be more explicit about the challenges that are particular to their method. Furthermore, for the camera ready the authors should use the additional page to carefully take into account the feedback of the reviewers, make the diagrams more readable, and add clarifications where needed to anticipate the questions that came up.

**Award:**

No

---

### Decision · Program_Chairs · 2022-09-14

Accept